# Enhanced nonlinear interaction of polaritons via excitonic Rydberg states in monolayer WSe$_2$

Jie Gu[1,2], Valentin Walther[3], Lutz Waldecker [4], Daniel Rhodes[5], Archana Raja[6], James C. Hone [5], Tony F. Heinz [4,7], Stéphane Kéna-Cohen[8], Thomas Pohl[3] & Vinod M. Menon [1,2✉]

Strong optical nonlinearities play a central role in realizing quantum photonic technologies. Exciton-polaritons, which result from the hybridization of material excitations and cavity photons, are an attractive candidate to realize such nonlinearities. While the interaction between ground state excitons generates a notable optical nonlinearity, the strength of such interactions is generally not sufficient to reach the regime of quantum nonlinear optics. Excited states, however, feature enhanced interactions and therefore hold promise for accessing the quantum domain of single-photon nonlinearities. Here we demonstrate the formation of exciton-polaritons using excited excitonic states in monolayer tungsten diselenide (WSe$_2$) embedded in a microcavity. The realized excited-state polaritons exhibit an enhanced nonlinear response $\sim g_{pol-pol}^{2s} \sim 46.4 \pm 13.9\,\mu eV\mu m^2$ which is ∼4.6 times that for the ground-state exciton. The demonstration of enhanced nonlinear response from excited exciton-polaritons presents the potential of generating strong exciton-polariton interactions, a necessary building block for solid-state quantum photonic technologies.

[1] Department of Physics, City College of New York, New York, NY, USA. [2] Department of Physics, Graduate Center of the City University of New York (CUNY), New York, NY, USA. [3] Center for Complex Quantum Systems, Department of Physics and Astronomy, Aarhus University, Aarhus C, Denmark. [4] Department of Applied Physics, Stanford University, Stanford, CA, USA. [5] Department of Mechanical Engineering, Columbia University, New York, NY, USA. [6] Molecular Foundry, Lawrence Berkeley National Laboratory, Berkeley, CA, USA. [7] SLAC National Accelerator Laboratory, Menlo Park, CA, USA. [8] Department of Engineering Physics, École Polytechnique de Montréal, Montréal, Quebec, Canada. ✉email: vmenon@ccny.cuny.edu

Exciton-polaritons, quasiparticles arising from the strong coupling between cavity photons and excitons in semiconductors, allow for the observation of exotic physical phenomena such as condensation[1–3], superfluidity[4], and quantized vortices[5], can be engineered to emulate systems such as atomic lattices for potential applications as quantum simulators[6], as well as for optoelectronic applications such as low energy switches[7], transistors[8], and interferometers[9]. This array of rich physical phenomena and the associated applications stem from the half-light half-matter make up of these quasiparticles. The photonic component lends the properties such as long-range propagation, small effective mass, and spatial coherence while the matter (exciton) component provides the interactions, spin-selectivity and nonlinearity[10–14]. The strength of this interaction depends on the fraction of the excitonic component present in the polaritons. In inorganic semiconductors that host Wannier–Mott excitons which can be described by a hydrogenic model, where the hole plays the role of the proton, the interaction strength is proportional to the exciton binding energy and the square of the exciton Bohr radius[12,15]. Systems such as monolayer transition metal dichalcogenides (TMDs) that have large ground state (1 s) exciton binding energies and also reasonable exciton radius (1 nm) have therefore become attractive platforms for exploring polariton physics and devices at elevated temperatures[16–21]. However, even in TMDs the exciton Bohr radius has been an impediment to realizing large exciton–exciton interaction strengths for the 1 s excitons[22,23]. This suggests the use of excited-state excitons as a potential approach to enhance polariton interactions and optical nonlinearities[24] by exploiting their larger Bohr radius.

Excited states of excitons have been studied in variety of materials such as $Cu_2O$, GaAs, TMDs, and halide perovskites[25–34]. Strong coupling of cavity photons to excited exciton states has been demonstrated in the GaAs and the perovskite systems recently[35,36]. However, the main motivation to go to excited states—to enhance the interaction strength—is yet to be convincingly demonstrated in any system. Here, we report the realization of strong cavity coupling to excited-state excitons and the formation of 2 s exciton polaritons in an archetypical two-dimensional TMD, $WSe_2$, embedded in a monolithic microcavity. We provide the experimental evidence for strong nonlinear interaction arising from the larger Bohr radius excited state excitons resulting in a blockade-like process. The associated interaction strength of 2 s exciton-polaritons is shown to be ~4.6 times larger than that of the 1 s exciton-polaritons in similar TMD systems, an enhancement that agrees with the scaling of the Bohr radius[37].

## Results

**Reflection spectrum of $WSe_2$ on DBR**. The sample structure is shown in Fig. 1a, where a 12-period silicon nitride (SiNx)/ silicon dioxide ($SiO_2$) distributed Bragg reflector (DBR) was first grown on silicon substrate via plasma enhanced chemical vapor deposition. Three layers of hexagonal boron nitride (hBN) encapsulated $WSe_2$ were then transferred on top of the DBR followed by spin coating 212 nm Poly (methyl methacrylate) (PMMA). A 40 nm silver layer was deposited to form the top mirror of the cavity. A similar approach was used to fabricate the $WS_2$ monolayer embedded cavity to study the 1 s excitons. The $WS_2$ monolayer was also encapsulated between the hBN layers and the cavity consisted of a similar bottom DBR mirror as used above for the $WSe_2$ cavity. The details of the devices' information can be found in "Methods", Supplymentary Fig. 3, and Supplymentary Fig. 4. Three layers of $WSe_2$ were used in these experiments due to the weaker oscillator strength (5–10 times) of the excited exciton state (2 s) compared with the ground state exciton (1 s) in TMDs (Supplementary Fig. 5). Typically, the Rabi splitting for monolayer 1 s exciton polaritons in TMDs is on the order of few tens of meV[38]. Since the total exciton-photon interaction strength is proportional to the square root of number of monolayers[21], more layers can be used to reach a larger Rabi splitting for the 2 s states. Figure 1b is an optical microscope image showing the three layers of $WSe_2$ after transfer onto the bottom DBR. To confirm the observation of the 2 s state, we

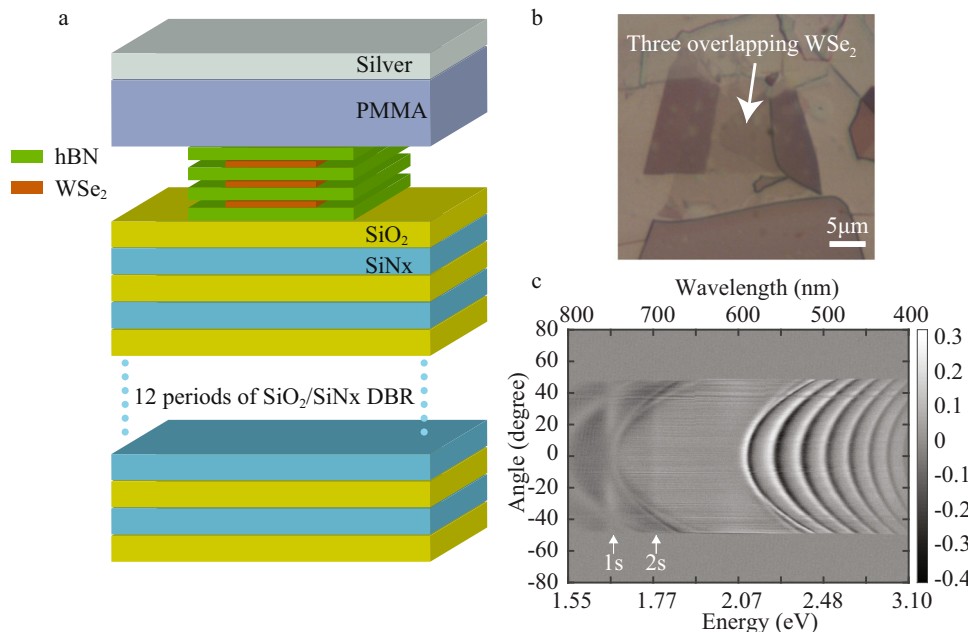

**Fig. 1 Reflection spectrum of $WSe_2$ on DBR. a** Schematic illustration of the sample structure. **b** Optical image after the multilayer hBN-$WSe_2$ stacking was transferred on top of the DBR. The area where three $WSe_2$ layers overlap is marked with the white arrow. **c** Angle resolved differential reflection of multilayer structure on DBR at room temperature. Absorption features can be observed at the energies of the 1 s and 2 s exciton and are marked by the white arrows.

performed a room temperature angle resolved white light reflection contrast measurement directly after the multilayer stack was transferred onto the bottom DBR. The reflection contrast is defined as $1 - R_{sample}/R_{ref}$, where $R_{sample}$ and $R_{ref}$ are reflected intensities from the sample area and a bare DBR area, respectively. Here a positive contrast indicates absorption in the sample. Both 1 s exciton (1.655 eV) and 2 s excitonic (1.780 eV) absorption features can be seen in Fig. 1c and their spectral positions are labeled by the white arrows. The parabolic features in Fig. 1c arise from the DBR side band.

**Temperauture dependent k-space reflection spectra.** Fourier space (k-space) imaging was carried out to determine the dispersion of the strongly coupled states. Figure 2a–c shows k-space white light reflection at different temperatures (140 K, 77 K, 15 K). The white solid lines are fits simulated using transfer matrix method. The white dashed lines represent the bare 2 s exciton energy ($E_{exc,2s}$) and bare cavity dispersions ($E_{cav}$). At 140 K (Fig. 2a), the 2 s exciton energy lies below the cavity resonance with a positive detuning of 11 meV and does not show strong coupling. Here detuning is defined as $\delta = E_{cav} - E_{exc,2s}$ at the angle of zero degree. The exciton energy blue shifts with decreasing temperature (Supplementary Fig. 5). By lowering the temperature (Fig. 2b and c), we can then blue shift the exciton energy across the cavity resonance leading to strong exciton-photon hybridization and clear anticrossing (Fig. 2d) between the upper and lower polariton branches with a measured Rabi splitting of $\Omega_0 \approx 7.7$ meV.

**White light intensity dependent k-space reflection spectra.** In order to investigate the nonlinear optical response of the excited exciton states, we measured the cavity reflection spectrum for varying intensities of the incident white light (see Methods for details on experiment). Upon increasing the intensity, the energy splitting between the polariton branches gradually decreases as shown in Fig. 3. Similar behavior was observed using the 1 s exciton ground state in WS$_2$ (Supplementary Fig. 6). Note that this behavior differs qualitatively from the effects of an increasing temperature shown in Fig. 2, such that we can exclude white light heating as a potential source for the observed intensity dependence of the cavity response.

Increasing the intensity can cause both a decrease in Rabi splitting[39–41] as well as an overall blue shift[10,11] of both polariton branches. The former effect dominates widebandgap materials[42] as well as organic systems[43] with strong binding energy while the latter effect is usually observed in low binding energy materials such as GaAs. In the limit of strong coupling both of these effects contribute to the overall nonlinear response. In the present experiments with TMDs having large exciton binding energies, the fomer effect dominates the nonlinear response as has been seen in the case of trion-polaritons[44,45] as well as for the ground state excitons[46]. We can understand and describe this behavior quantitatively in terms of phase space filling in momentum space or polariton blockade in real space, whereby the interaction between polaritons prevents their simultaneous generation within a blockade radius $R_{bl}$ and thereby inhibits the formation of polaritons within the distance $R_{bl}$. Analogous to the emergence of optical nonlinearities in cold atomic Rydberg gases[47–49], this

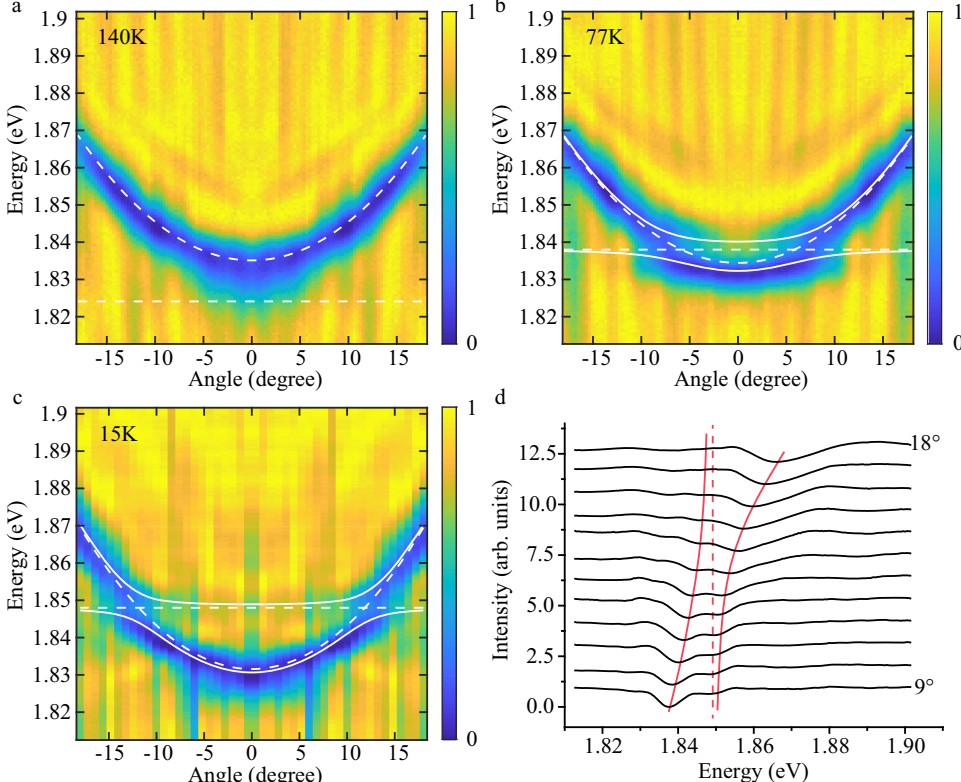

**Fig. 2 Temperature dependent cavity coupling to excited-state excitons. a–c** Angle resolved reflection at different temperatures of 140, 77, and 15 K. The dashed lines indicate the bare cavity and exciton dispersions, while the solid lines correspond to the resulting dispersion relations of the two polaritons induced by the strong coupling between the excitons and the cavity photons. **d** Line cut from the 15 K data between $9°$ and $18°$. Those data have $y$ axis offset for clarity. A clear anti-crossing is observed (red trend lines). The extracted Rabi splitting is 7.7 meV. The dashed line shows the 2 s exciton energy at 1.848 eV.

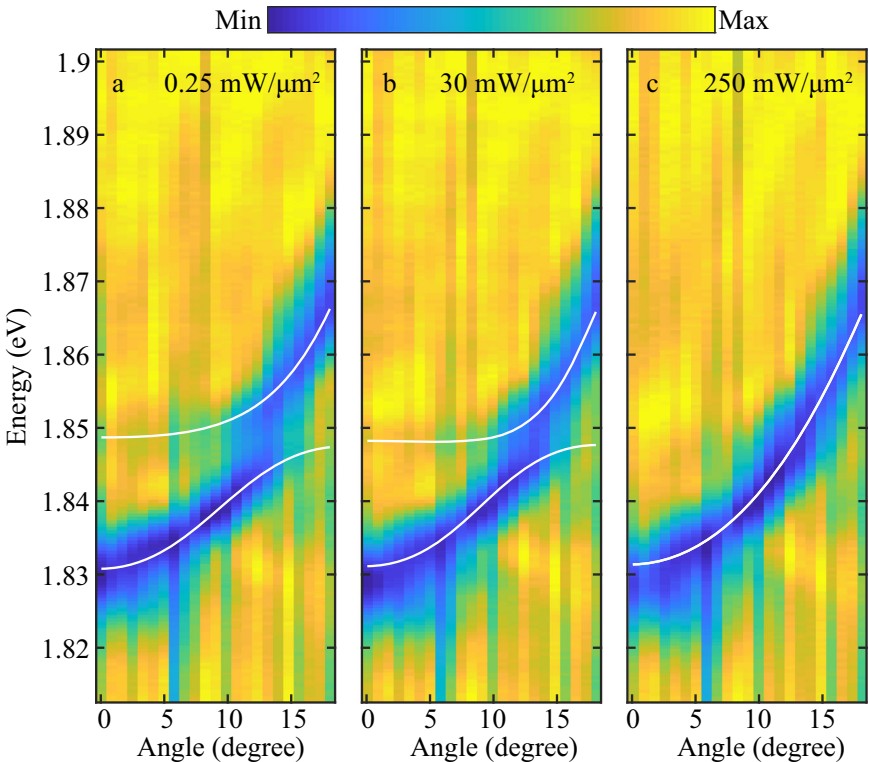

**Fig. 3 White light intensity dependent cavity reflection. a–c** Angle resolved reflection for different incident intensities of 0.25 mW/μm² (**a**), 30 mW/μm² (**b**), and 250 mW/μm² (**c**). Higher intensities imply a higher density of polaritons, which eventually leads to an excitation blockade induced by the strong interaction between excitons. This polariton blockade reduces the effective coupling to the cavity, observed as a decreasing splitting of the two polariton branches in our experiments. The analysis for nonlinear interacion is carried out below 30 mW/μm² where the system is well under the strong coupling regime.

mechanism gives rise to a nonlocal optical nonlinearity with a nonlinear kernel (Supplementary Note 2)

$$\chi^{(3)}(r) = \frac{16g^4}{|\Gamma|^2\Gamma}\frac{iU(r)}{\Gamma + iU(r)}, \qquad (1)$$

where $g$ is the exciton-photon coupling strength, $U(r)$ is the interaction between two excitons at a distance $r$ and $\Gamma = \gamma - 2i\triangle$ is determined by the width $\gamma$ of the exciton resonance (2 meV for both 1 s and 2 s)[34,50] and $\triangle$, which is the energy difference between the incident light and the exciton resonance (Supplementary Note 2). We choose here a real-space formulation of the interacting exciton problem that is better suited to illustrate the scaling with the principle quantum number. The potential can thus, incorporate the exact exciton–exciton potential, as is approximated by the direct, exciton exchange, electron exchange, and hole exchange[51]. For low-lying states the dominant contribution is that of exchange capturing the Pauli blockade of identical Fermions. As was pointed out in[52], the real space interpretation of this blockade is an excluded area around each exciton, in which no further excitation can take place. This process is exactly and transparently captured by an effective hard-core repulsive exciton–exciton interaction $U(r)$ with a length scale on the order of the excitonic Bohr radius. This nonlocal form of the optical response describes both a polariton blockade within a blockade radius, $R_{bl}$, determined by $U(R_{bl}) = |\Gamma|$, as well as a cavity line shift that arises from exciton–exciton interactions outside of $R_{bl}$. Considering a situation in which the interaction predominantly leads to a phase space filling (real space blockade), one can obtain a simple expression (Supplementary Note 2) for the resulting nonlinear effect on the Rabi splitting $\Omega$ as a function of the polariton density

$$\Omega(n) = 2g\left(1 - \pi R_{bl}^2 n/2\right) \qquad (2)$$

in terms of the blockade radius $R_{bl}$ and the polariton density $n$. Equation (2) affords a simple geometrical interpretation, indicating that the two polariton branches shift towards each other with the number of blocked polaritons, as given by the product of the polariton density and the blockade area $\pi R_{bl}^2$. In this simple picture, a Rydberg exciton reduces the material area available for further creation of excitons. The origin of this excitation blockade here is the Pauli exclusion principle, preventing the simulatanous excitation of two exciton within a radius on the order of the exciton size. The same effect can also be understood in reciprocal space as phase-space filling, where electrons and holes occupy the available states in the conduction and valence bands, thus reducing the oscillator strength of the corresponding exciton states[40,41,44]. We note that the correspondingly decreasing splitting of the two polariton resonances, while remaining in the strong coupling regime, indicates the inhibition of polariton formation within a distance $R_{bl}$ and eventually saturates the exciton density at $\sim 1/(\pi R_{bl}^2)$ with increasing intensity. As mentioned above, exciton interactions naturally cause a shifting as well as a phase space filling effect. The relative extent of both effects depends on the value of the blockade radius –suppressing exciton generation and decreasing the Rabi splitting (phase space filling) – and the contribution of the residual interaction potential outside of $R_{bl}$ – causing an overall line shift of both polaritons. This observation, in turn, also suggests a method of separating the contributions of the polariton blueshift and the closing of the Rabi splitting in a unified picture: the very strong interactions at short distances break the coupling to the cavity light mode,

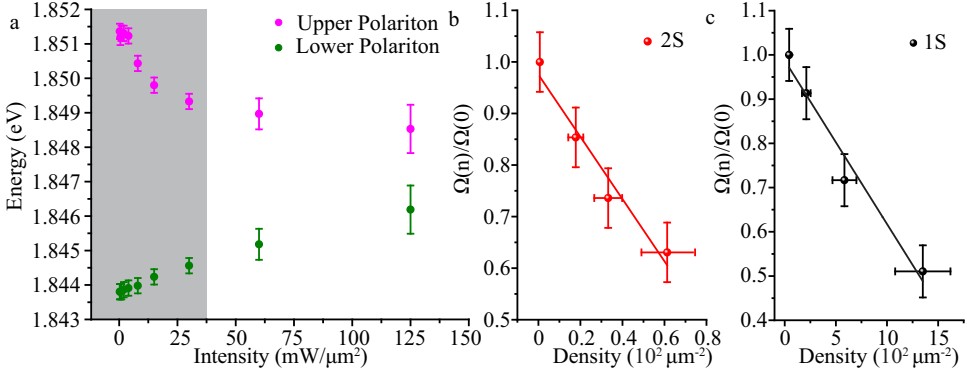

**Fig. 4 Interaction induced nonlinear cavity response. a** Energies of the upper polariton and lower polariton under different injected intensity at the Rabi splitting angle where exciton and photon component have the same fraction. The shaded region corresponds to the strong coupling regime used to estimate the strength of the nonlinearity. **b**, **c** Normalized Rabi splitting as a function of the polariton density for 2 s (**b**) and 1 s (**c**). The solid lines show fits of Eq. (2) to our low-density data, yielding blockade radii of 61.9 ± 9.3 nm and 15.3 ± 3.2 nm for the 2 s- and 1 s-exciton, respectively. The errors bars represent one standard deviation from fitting the reflectivity at each input power.

leading to the described mechanism of blockade. Outside this blockade region, but at distances of non-negligible exciton–exciton interactions, a disperive interaction leads to a blue shift of both polariton branches. The transition between these two regimes is determined by the interaction potential $U(r)$ and the detailed dephasing and decay rates of the excitons. Excited exciton states, thus offer a systematic way to increase the interaction range and thereby enhance the former effect as the blockade radius scales with the Bohr radius and hence with the principle quantum number, as we will demonstrate below.

**Interaction induced nonlinear cavity response.** Figure 4a shows the energy of both polariton branches for the 2 s state at their avoided crossing as a function of the input intensity (1 s data is shown in Supplementary Fig. 6). Although the polariton splitting vanishes at the highest intensities, we only consider intensities below 30 mW/μm², where the system is well in the strong-coupling regime to quantitatively extract the strength of the nonlinearity. By subtracting the lower polariton energy from the upper polariton energy, we obtain the density-dependent Rabi splitting, which is shown in Fig. 4b and c for the excited 2s-exciton and the ground state 1s-exciton, respectively. The observed Rabi splitting indeed decreases linearly for small intensities regime as predicted by Eq. (2). In this regime, we can use the known linear relation between the pump intensity and the polariton density (see Supplementary Note 3) to fit the relation Eq. (2) to our measurements, as shown in Fig. 4b and c. This in turn allows to determine the polariton blockade radius, and yields $R_{bl}^{(2s)} = 61.9 \pm 9.3$ nm for the 2 s exciton state corresponding to $g_{pol-pol}^{2s} \sim 46.4 \pm 13.9$ μeVμm². In contrast, our analogous measurements for 1 s ground state excitons in WS₂ reveal a weaker nonlinearity, $g_{pol-pol}^{1s} \sim 10.0 \pm 4.2$ μeVμm² and an associated blockade radius of $R_{bl}^{(1s)} = 15.3 \pm 3.2$ nm. In fact, the exciton radii have been measured experimentally[32,53] to be $a_{1s} = 1.8$ nm for the 1 s state in WS₂ and $a_{2s} = 6.6$ nm for the 2 s state in WSe₂, respectively. Therefore, the ratio $a_{2s}/a_{1s} = 3.66$ is indeed close to the enhancement $R_{bl}^{(2s)}/R_{bl}^{(1s)} = 4.0 \pm 1.0$ of the blockade radius deduced from our measurements. For phase space filling induced nonlinearity, the polariton–polariton interaction strength is proportional to the square of the Bohr radius and the Rabi splitting, $g_{pol-pol} \propto a_B^2 \hbar\Omega$[37]. This gives an estimated ratio of $g_{pol-pol}^{2s}/g_{pol-pol}^{1s} = 3.8$, where the experimental Rabi splitting for the 2 s (7.7 meV) and 1 s (27 meV) have been used. This

estimated ratio based on the above expression is in reasonable agreement with the ratio obatined experimentally of $g_{pol-pol}^{2s}/g_{pol-pol}^{1s} = 4.6 \pm 2.3$.

**Discussion**

The enhancement of the exciton blockade radius demonstrated in our experiments corresponds to a 4.6 times enhancement of the polariton–polariton interaction strength. This observed growth of the nonlinearity agrees with the expected scaling of the resulting effective polariton–polariton interaction, due to the increasing range of the exciton interaction with their principal quantum number $n$. This result presents a systematic approach towards the enhancement of polaritonic nonlinearities with the ultimate goal of reaching the regime of nonclassical light[13,14] and photon blockade that is reached when a single photon can block all further excitations in the cavity. Realizing stronger cavity coupling and narrower line width combined with higher-lying exciton states (>3 s) would yield larger blockade radii and permit approaching the photon-blockade regime. At high principle quantum numbers, long-ranged direct dipole–dipole interactions are expected to contribute most to the overall exciton interactions, leading to an even stronger scaling with the principle quantum number[24]. In this regime, additional interaction contributions from the excitons in different layers may lead to an even stronger scaling with $n$.

Such an enhancement with the increasing excitonic excitation levels together with the demonstrated realization of excited-state exciton-polaritons opens up a promising approach to reach the regime of strong polariton interactions in semiconductor microcavities. In the long term, such enhanced nonlinearities could serve as a mechanism for the generation of nonclassical light and ultimately strongly correlated few-photon quantum states with integrated microcavities.

**Methods**

**Sample fabrication.** The cavity consists of van der Waals stack of [hBN-WSe₂] x 3 sandwiched between a 12-period bottom SiNx/SiO₂ DBR, and top Poly (methyl methacrylate) (PMMA) layer and silver mirror as shown in Fig. 1a. The DBR was grown by plasma-enhanced chemical vapor deposition (PECVD) on a silicon substrate using a combination of nitrous oxide, silane, and ammonia at a temperature of 350 °C. Each SiO₂ and SiNx layer is 113 nm, 82.5 nm, with refractive indices 1.46 and 2.03, respectively. The WSe₂ monolayers were obtained via direct exfoliation from high-quality WSe₂ crystal grown by vapor flux method. The hBN capping layers were obtained through exfoliation from commercially purchased hBN crystal (HQ Graphene). Monolayer WSe₂ and different layers of hBN were identified under an optical microscope using their different contrasts on SiO₂/Si substrate. The hBN/WSe₂/hBN/WSe₂/hBN/WSe₂/hBN stack was realized by following the poly-propylene carbonate (PPC) pick up technique[54]. The thickness of

each layer can be found in Supplementary Fig. 3. The completed stack was transferred onto the bottom DBR at 120 °C followed by soaking the entire sample in chloroform for 2 h to remove PPC residue. 212 nm thick PMMA (495 A4 from Michrochem, 2020 rpm for 50 sec) was spin coated onto the hBN-WSe$_2$ stack to have the 3λ/2 cavity mode in resonance with exciton energy. After the spin coating, the sample was heated for 2 mins to dry the PMMA. Finally, a 40 nm thick silver top mirror was deposited using electron beam evaporation. A similar approach was used to fabricate the WS$_2$ monolayer embedded cavity. The WS$_2$ monolayer was also encapsulated between the hBN layers and the cavity consisted of a similar bottom DBR mirror as used above for the WSe$_2$ cavity (details of the WS$_2$ sample can be found in Supplymentary Fig. 4). In both cases we use a 3λ/2 thick cavity because it's difficult to spin coat a uniform PMMA layer that is thinner than the van der Waals stack structure (~60 nm). Hence, we used a thicker PMMA capping layer, resulting in a 3λ/2 thick cavity. The mode distributions are shown in Supplementary Fig. 3 and Supplementary Fig. 4 for WSe$_2$ and WS$_2$ cavity structures, respectively, where one can see that the monolayers are placed near the second maxima of the cavity field. These heterostructure samples have inhomogenities arising from air bubbles trapped during the fabrication process. However, we have carried out our measurements using a laser spot size (1 μm$^2$) that is far smaller than the uniform area on the sample (~20 μm$^2$) as shown by the white dashed circle in Supplementary Fig. 2.

We have studied the 1 s state of WS$_2$ and 2 s state of WSe$_2$, since they lie in a similar spectral range and therefore permit to use nearly identical cavity structures, which minimizes potential effects that may otherwise arise from different cavity fabrication procedures for the two exciton states.

**Optical measurements**. The angle-dependent reflection measurements were taken using Fourier space imaging technique. A super continuum pulsed light source (NKT Photoincs, repetition rate 1.98–80 MHz, pulse duration 20 ps) passing through a long pass filter at 650 nm (550 nm) and a short pass filter at 750 nm (650 nm) was used to selectively excite the 2 s (1 s) exciton state in WSe$_2$ (WS$_2$). The single pulse averaged power was used in the maintext. The setup is coupled with Princeton Instruments monochromator with a PIXIS: 256 EMCCD camera. The spot size of the laser was 1 μm$^2$ and ensured that we were probing a uniform area of the sample (~20 μm$^2$).

## Data availability
Data are available from the authors upon reasonable request.

## Code availability
Code are available from the authors upon reasonable request.

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

## Acknowledgements

Work at City College of New York—CUNY was supported by the National Science Foundation through the MRSEC program DMR—2011738 (J.G), and the ARO MURI program (W911NF-17-1-0312) (V.M.M). Sample growth at Columbia was supported by MRSEC program DMR—2011738. Spectroscopy studies at Stanford were supported by the National Science Foundation through grant DMR-1708457, by the Gordon and Betty Moore Foundation's EPiQS Initiative through grant GBMF9462, and by a fellowship from the Alexander von Humboldt Foundation (LW). A.R. gratefully acknowledges support through the Laboratory Directed Research and Development Program of Lawrence Berkeley National Laboratory under U.S. Department of Energy Contract No. DE-AC02-05CH11231. S.K.C. acknowledges support from the NSERC Discovery Grant Program and the Canada Research Chairs. Work at Aarhus was supported by the Carlsberg Foundation through the "Semper Ardens" Research Project QCooL, by the DFG through the SPP1929, by the European Commission through the H2020-FETOPEN project ErBeStA (No. 800942), and by the Danish National Research Foundation through the Center of Excellence "CCQ" (Grant agreement no.: DNRF156). The authors also acknowledge the use of the Nanofabrication Facility at the CUNY Advanced Science Research Center for the fabrication of the devices.

## Author contributions

V.M.M., T.F.H., J.G. conceived the experiments. J.G. fabricated the devices. D.R., J.C.H. grew the bulk WSe$_2$ crystal. J.G., L.W., performed the measurements with input from A.R. J.G. performed data analysis. T.P., S.K.C., V.W. did the theoretical modeling. V.M.M supervised the project. All authors contributed to write the paper and discuss the results.

## Competing interests

The authors declare no competing interests.
