## [Peer Review File · Nature Communications]

REVIEWER COMMENTS

Reviewer #1 (Remarks to the Author):

In the manuscript, authors explore the nonlinear optical response of the exciton-polaritons, formed by the first excited exciton state (2s) in transition metal dichalcogenide monolayer. They show that the main mechanism of the nonlinearity corresponds to the Rabi splitting saturation which the authors term "exciton blockade". Moreover, authors demonstrate that the 2s exciton state corresponds to larger nonlinearity as compared to the ground 1s state scaling with the exciton main quantum number according to theoretical predictions.

The paper presents novel and important experimental results and could in principle be accepted as a publication in Nature Communications. However, there are a couple of questions, which I feel should be addressed before the manuscript could be accepted.

1. I have doubts with the validity of the method of calculating the polariton density presented in Supplementary (S6). According to this approach, in a hypothetical non-absorbing cavity, the density of polaritons will be exactly zero, since no power is absorbed. But this definitely is not the case, since the polariton density will be just proportional to the intensity of the cavity mode inside the cavity, which is non-zero even in the case of the absorption-free cavity.

2. I failed to find the comparison of the absolute values of the polariton nonlinearity of 1s and 2s excitons with the previously reported values. While authors cite some of the papers where the polariton nonlinearity has been directly obtained - (e.g. Nature Nanotech. 13, 906-909 (2018)) - no direct comparison is provided. It is also worth adding some recent papers where the nonlinear response of TMD exciton-polaritons [Light: Science & Applications 9, 56 (2020)] and trion-polaritons [Nature Communications 11, 3589 (2020)] was measured and compare the obtained values of nonlinearity to the previously reported.

Reviewer #2 (Remarks to the Author):

The paper by Gu et al reports a study on the exciton polaritons forming by highly excited excitonic states of TMDC material. The authors observed enhanced polariton-polariton interaction for the excited exciton polariton compared with its ground state counterpart. The authors attributed the stronger nonlinear response to the larger exciton radius of the Rydberg excitonic states. Microcavity polaritons are of particular interesting on a plethora of emerging technological applications. The paper exemplifies enhanced intrinsic non-linearities in strong coupled light-matter interaction. It is potential interested for the community. Overall, the paper is well organized and clearly written. I would support its publication if the authors could address the following technical questions.

My major concern is about the microscopic mechanism. This manuscript lacks of vision about the implications of their discovery: I would like to see an in-depth explanation for the physics of enhanced polariton-polariton interactions (or equally, the enlarged exciton blockade radius) of excited excitonic states.

Here is also a few smaller questions:

1. As it is known, distinctively different from ground state, the wave-function of 2S excitonic state is not centro-symmetric. The definition of polariton size is thus not isotropic for the high Rydberg states. The author should discuss the distorted geometry does not seriously affect polariton-polariton interaction nor alters the physics behind equation 2.

2. Would the inter-layer excitonic effect influence the results? Considering the vertical stacking

distance is much smaller than the extracted polariton-polariton interaction distance.

3. Could the author comment on the possible applications? Such as light-emission or coherent light sources.

Reviewer #3 (Remarks to the Author):

Authors present very interesting study of the reduction of the exciton-photon coupling strength (Rabi energy) for monolayers of WSe₂ and WS₂ placed in an optical cavity. The polariton modes are observed for both 1s and 2s excited exciton states. Authors use high density broad energy excitation (super continuum pulsed light source), resonant to upper and lower polariton modes. Upon increased excitation density the reduction of the Rabi splitting is observed.

I find the experiment of importance as it clearly demonstrates the excitonic saturation density detected at both 1s and 2s states in 2D crystals, which is very valuable. The interpretation of the observation is however not very accurate as authors attribute the effects to „exciton blockade induced by strong interactions between excitons“.

Authors do not cite the experiments demonstrating the same effect i.e. the quenching of the Rabi oscillations at high exciton density, close to saturation limit, in polariton system. A. Huyhn et al. [Physica E 13, 427 (2002)] already discussed reduction of the Rabi energy within the effects of Pauli blocking (phase-space filling) and excitonic saturation, similar discussion can be found in M. Saba et al. Physica B 272, 472 (1999), or very recently in R. P. A. Emmanuele Nature Communications 11, 3589 (2020).

In present manuscript the authors not clearly address different regimes of interactions with respect to polariton density present in polariton system. The introduction refers to „quantum nonlinear optics“ and „single-photon non-linearities“ but also to „polariton condensation“, „superfluidity“ and „vortices“ - the coherent bosonic phenomena, where the interactions are dominated by Coulomb exchange interactions (i.e. exciton-exciton scattering), and the situation does not correspond to their experiment, where the system is driven to excitonic saturation density, where the strong coupling regime is lost.

I hope that the authors do not tend to force a new interpretation on already known effects with a new „exciton blockade“ which is not precise enough. The observed effects can be interpreted within the quenching of the Rabi splitting induced by the saturation of molecular optical transitions due to states filling at higher excitation densities (Pauli blocking), the saturation of exciton oscillator strength.

Equations (1-2) refer to the same effect of the non-linear change of the refractive index due to the saturation of the excitonic transition, where at high densities the occupied states do not take part in the absorption process and reduce Rabi splitting. Authors refer to the „close analogy between the optical non-linearities in cold atomic Rydberg gases“ but the reference is required as this analogy is not evident. What's more the atomic Rydberg gases in cavities (Rabi oscillations) differ much from the polariton system (vacuum field Rabi oscillations).

Despite the above criticism, I have to admit that the results are very valuable from the point of view of recent publications. In my opinion the authors should put more emphasis on the observed saturation densities which are relatively low. With comparison to V. Kravtsov et al. Light Sci. Appl. 9, 56 (2020) where the Mott transition is claimed to be at $10^{14}/\text{cm}^2$ and the polariton density at $10^{12}/\text{cm}^2$, in the manuscript authors have the densities up to at $10^{11}/\text{cm}^2$ and already observe the Rabi quenching. This demonstrate huge discrepancy in the literature.

Summarizing my remarks, the novelty of the research is not very high:

- the observation of exciton polaritons in 2s state in the absorption process in 2D materials is not new [M. Krol et al 2020 2D Mater. 7 015006]
- the model is also widely used to describe the saturation of molecular optical transitions due to states filling at higher excitation densities
- the idea of admixture of higher excited states to polariton modes to increase the non-linear effects was also demonstrated [V. Walther Nature Communications 9, 1309 (2018); V. Kravtsov et al. Light Sci. Appl. 9, 56 (2020)], also with 2p that have a dipole moment [Nano Lett. 2020, 20, 1676–1685]
- the interpretation of the observed effects of the quenching of Rabi splitting should be corrected and the different types of interactions in polariton system should be clarified and properly addressed.

The new experimental finding presented in the manuscript is that the strong coupling at 2s state is lost at lower densities than at 1s state. Even though I think that this result is very valuable and I would like to see it published, I do not recommend the manuscript for the publication in Nature Communications.

Response to Reviewer 1

We thank the reviewer for the detailed assessment of our manuscript and the thoughtful comments on our work. We are particularly happy about the reviewers' positive outlook on our results, finding that our "*paper presents novel and important experimental results*" and that is suitable for publication in Nature Communications. The reviewer has raised a couple of interesting questions which we have all addressed as described below:

1. "*I have doubts with the validity of the method of caculcating the polariton density presented in Supplimentary (S6). According to this approach, in a hypothetic non-absorbing cavity, the density of polaritons will be exactly zero, since no power is absorbed. But this definitely is not the case, since the polariton density will be just proportional to the intensity of the cavity mode inside the cavity, which is non-zero even in the case of the absorption-free cavity.*"

We thank the reviewer for bringing this up this important comment. In our original approach, "absorption" in the SI should be understood as simply meaning the dip contrast, which was somewhat confusing. This yields the same result as that obtained using input-output theory. However, that approach does not account for the fact that part of the contribution to the photonic linewidth comes from absorption in the metal mirror. We have modified this section (S6) to exactly account for the radiative part of the photon linewidth. Specifically, a detailed calculation of the coupling constant κ properly accounting for the DBR penetration depth and metal refractive index gives $4.3 \times 10^{12} \text{ rad/s}$ (**SI Section S6**).

2. "*I failed to find the comparison of the absolute values of the polariton nonlinearity of 1s and 2s excitons with the previously reported values. While authors cite some of the papers where the polariton nonlinearity has been directly obtained - (e.g. Nature Nanotech. 13, 906-909 (2018)) - no direct comparison is provided. It is also worth adding some recent papers where the nonlinear response of TMD exciton-polaritons [Light: Science & Applications 9, 56 (2020)] and trion-polaritons [Nature Communications 11, 3589 (2020)] was measured and compare the obtained values of nonlinearity to the previously reported.*"

This is an excellent suggestion. In our experiment, we obtain polariton nonlinearity of $g_{pol-pol}^{2s} \sim 46.4 \pm 13.9 \mu\text{eV} \mu\text{m}^2$ for the 2s-exciton cavity and $g_{pol-pol}^{1s} \sim 10.0 \pm 4.2 \mu\text{eV} \mu\text{m}^2$ for the 1s-exciton measurements. The latter is comparable to the value of $\sim 7 \mu\text{eV} \mu\text{m}^2$ reported in [Stepanov et al. ArXiv 2007.00431v1 (2020)]. This value for the 1s state is higher than what has been recently reported in two other works [Light: Science & Applications 9, 56 (2020)] and [Nature Communications 11, 3589 (2020)] for the 1s exciton-polariton. The main result of the present work, showing the scaling of the polariton-polariton interaction strength by using excitonic Rydberg states (increasing with principle quantum number) is in agreement with what is expected for a phase space filling (PSF) type nonlinear response which is proportional to the Rabi splitting and square of the Bohr radius: $g_{pol-pol} \propto a_B^2 \hbar \Omega$. The estimated ratio of $g_{pol-pol}^{2s} / g_{pol-pol}^{1s} = 3.8$, where the experimental Rabi splitting for the 2s (7.7meV) and 1s (27 meV) have been used. This estimated ratio based on the above expression is close to the ratio obtained experimentally of $g_{pol-pol}^{2s} / g_{pol-pol}^{1s} = 4.6 \pm 2.3$. This is now included in the revised manuscript (**Page 11**).

We would like to refrain from a direct comparison with the measurements of [*Nature Nanotech.* *13*, 906-909 (2018)] since this work did not use a cavity and used PL to probe propagating surface polaritons, which is quite different from the current study. Similarly, trion nonlinearities do not seem to afford a direct meaningful comparison, since in this case the nonlinearity arises from a finite preset electron density that seeds trion formation, whereby a smaller electron density speeds up saturation and therefore leads to a larger saturation.

To the best of our knowledge, the measured 2s-nonlinearity represents one of the largest polaritonic nonlinearity reported in TMD materials to date, and larger values will be possible with a stronger cavity coupling and higher lying exciton states. For example, the 3s-state would already yield a two-order of magnitude enhancement compared to the exciton ground state. We thank the reviewer for this suggestion and have now included a discussion of the estimated polariton nonlinearities (**pages 10, 11 in revised manuscript**).

Response to Reviewer 2

We thank the reviewer for the thorough assessment of our work and the positive feedback, highlighting the general importance for “*emerging technological applications*” and the high interest in the field. We are happy that the reviewer supports publication of our manuscript and appreciate the helpful comments, which have all been addressed as described below:

1. “*My major concern is about the microscopic mechanism. This manuscript lacks of vision about the implications of their discovery: I would like to see an in-depth explanation for the physics of enhanced polariton-polariton interactions (or equally, the enlarged exciton blockade radius) of excited excitonic states.*”

We thank the reviewer for this remark and have now included a dedicated description of the underlying interaction mechanisms in the revised manuscript, along with a discussion of their enhancement for excited exciton states. In short, higher excited states have a larger exciton radius, scaling as n^2 with the principal quantum number. This implies a larger distance at which exchange interactions affect their optical generation (scaling as n^2). The increased exciton size also implies a larger polarizability of the excitonic wave function and, therefore, an enhanced electrostatic interaction, which at large distances would also lead to an approximately quadratic scaling of the blockade radius [see e.g. *Nat. Commun.* **9**, 1309 (2018)]. While their relative importance for the described blockade effect depends on other parameters, such as the exciton linewidth, both contributions to the exciton interaction yield a similar scaling, n^2 , with the principal quantum number, which is well demonstrated by our experiments.

The perspective of our work, is therefore to lay out a systematic approach to enhancing nonlinearities of semiconductor polaritons, whereby the discussed exciton blockade may be used to generate nonclassical light, and its enhancement (for larger n , stronger photon coupling g , and/or smaller exciton linewidths γ) provides an essential step into the photon blockade regime. We

thank the reviewer for pointing out that these important points can be described better and have now included corresponding discussions in the revised version of the manuscript (**Page 11**).

2. *“As it is known, distinctively different from ground state, the wave-function of 2S excitonic state is not centro-symmetric. The definition of polariton size is thus not isotropic for the high Rydberg states. The author should discuss the distorted geometry does not seriously affect polariton-polariton interaction nor alters the physics behind equation 2.”*

This is a very good comment. Indeed, in the simplest approximation, the series of s -states is characterized by a vanishing angular momentum. Therefore, the excitonic wavefunction of an ns -state is radially symmetric for any principal quantum number n , such that the interaction potential U in Eq.(1) has indeed radial symmetry and Eq.(2) is valid. Although the crystal lattice in TMDs can break the symmetry, it has been shown previously that the deviations are small [*Nature* **513**, 213 (2014)] and hence our assumption is reasonably valid.

We would like to note, however, that the underlying theory can be equally applied to non-symmetric potentials (see Supplementary Section S5), and for example be used to describe optical nonlinearities between excitonic p -states which have non-symmetric interactions, as pointed out by the reviewer (see Ref. *Phys. Rev. Lett.* **125**, 097401 (2020)).

3. *“Would the inter-layer excitonic effect influence the results? Considering the vertical stacking distance is much smaller than the extracted polariton-polariton interaction distance.”*

This is a very good question. The wave function of the excitons is confined to the two-dimensional plane of each layer sandwiched between hBN layers and there are no inter-layer excitons in our system. As this eliminates any wave function overlap for excitons in different layers, there should be no exchange interactions across the layers. Since the extracted interaction range R_{bl} still suggest that the major contributions to the nonlinearity are stemming from phase space filling nonlinearity for both $1s$ and $2s$ excitons, we do not expect significant effects of interlayer interactions.

Yet, excitons can in principle polarize each other and experience electrostatic interactions across the layers, and such effects may become important for higher excited exciton states, as the reviewer points out correctly.

4. *“Could the author comment on the possible applications? Such as light-emission or coherent light sources.”*

The enhanced interactions explored in our work suggest future applications in quantum photonics. Here, a stronger cavity coupling and narrower line width combined with higher-lying exciton states would yield larger blockade radii and permit to approach the photon-blockade regime. In the long term, such enhanced nonlinearities could serve as a mechanism for the generation of nonclassical light and ultimately strongly correlated few-photon quantum states with integrated microcavities. Following the referee’s suggestions these applications and perspectives are now discussed in the concluding section of the revised manuscript (**page 11 in revised manuscript**).

Response to Reviewer 3

We thank the reviewer for the detailed assessment of our manuscript and are glad about the reviewers positive conclusion about the relevance of our work, finding our manuscript to “*present very interesting study*”, and pointing out that “*the experiment [is] of importance*” and that its “*result is very valuable*”. The reviewer has raised some interesting questions about the novelty and interpretation of the reported findings, which have all been clarified entirely, as detailed below:

1. “*The interpretation of the observation is however not very accurate as authors attribute the effects to „exciton blockade induced by strong interactions between excitons”.*”

We appreciate this comment by the reviewer. The given expression for the leading-order nonlinearity follows from a rigorous derivation that starts from the basic evolution equations for excitons and cavity photons in the presence of arbitrary exciton interactions (described by an interaction potential $U(r)$). We hope that the reviewer can agree that the presented analysis provides an accurate and consistent description of the experiment. We would like to note that the underlying theoretical treatment has been successfully applied to a wide range of quantum optics experiments with atomic systems and provenly provides a highly accurate theoretical framework to describe polariton interactions in these settings. Most recently, the presented blockade picture and underlying theory was demonstrated to yield a highly accurate quantitative description of optical nonlinearities in Cu_2O semiconductors, induced by exciton interactions.

There is, however, a direct correspondence to other interpretations in terms of Pauli exclusion originating from the composite nature of the excitons [see e.g. Ciuti et al. *Semicond. Sci. Technol.* **18**, S279–S293 (2003)]. To lowest order (i.e. in the regime where the energy shifts scale linearly with the light intensity), it suffices to consider two excitons. Here it is well known that the short-range part of the interaction potential that arises from exchange (of the excitons’ electrons and holes) precisely describes the Pauli exclusion of the Fermionic constituents of the excitons. If we reduce the interaction potential $U(r)$ to pure exchange processes (i.e. neglecting all relevant electrostatic contributions) the presented theoretical treatment simply corresponds to a real-space formulation of the phase-space filling description [see e.g. Ciuti et al. *Semicond. Sci. Technol.* **18**, S279–S293 (2003); Schmitt-Rink et al. *Phys Rev. B* **32**, 6601 (1985)], which is usually argued in momentum space. Exciton suppression due to Pauli-exclusion (phase-space filling) is therefore contained in the present formulation, which accounts for all other relevant interaction terms (i.e. electrostatic), which are known to become important for higher excited states. Additional reasons for why we believe the described pictures offers additional value in the present context are discussed below in the answers to the reviewer’s further questions.

We thank the reviewer for pointing out that the described correspondence between these two pictures may not be obvious to general reader and have therefore added a corresponding clarifying discussion in the revised manuscript in order to avoid any potential confusion (pages 8, 9, 10 in revised manuscript).

2. “*Authors do not cite the experiments demonstrating the same effect i.e. the quenching of the Rabi oscillations at high exciton density, close to saturation limit, in polariton system. A. Huyhn*”

et al. [Physica E 13, 427 (2002)] already discussed reduction of the Rabi energy within the effects of Pauli blocking (phase-space filling) and excitonic saturation, similar discussion can be found in M. Saba et al. Physica B 272, 472 (1999), or very recently in R. P. A. Emmanuele Nature Communications 11, 3589 (2020).”

We have included the suggested references in the revised manuscript (Ref. 40, 41, 44 in revised manuscript). The original manuscript already contained a related discussion and corresponding reference, and the suggested additional references are included there.

3. *“In present manuscript the authors not clearly address different regimes of interactions with respect to polariton density present in polariton system. The introduction refers to „quantum nonlinear optics” and „single-photon non-linearities” but also to „polariton condensation”, „superfluidity” and „vortices” - the coherent bosonic phenomena, where the interactions are dominated by Coulomb exchange interactions (i.e. exciton-exciton scattering), and the situation does not correspond to their experiment, where the system is driven to excitonic saturation density, where the strong coupling regime is lost.”*

As we describe in the manuscript (see also Fig.4), we focus on the regime of low densities, where the polariton energies shift linearly with the laser intensity. In this regime, the optical nonlinearities clearly arise from binary interactions, as we discuss in the manuscript.

The referee is correct, that Coulomb exchange interactions are expected to yield the major contribution to collisional exciton interactions, that are, e.g., observed for polariton condensation in emission measurements (i.e. different from the current experiment). However, it is the very same fundamental interaction that gives rise to a reduction of the Rabi splitting due to Pauli exclusion, which is often described in momentum space and therefore understood as a phase space filling. This close connection appears since both effects stem from the Pauli exclusion of the Fermionic constituents of the excitons. Yet, our real-space treatment that leads to a blockade picture of the underlying process, reveals this connection more directly, revealing that both optical nonlinearities arise from exchange interactions between excitons (See also recent work by Kyriienko et al. *Phys. Rev. Lett.* **125**, 197402 (2020))

This real-space treatment and associated blockade picture make ensuing applications more apparent, since the underlying exciton blockade that tends to decrease the Rabi splitting for a classical coherent input state of the light, provides a clear path towards a photon blockade (e.g. with stronger cavity coupling) to achieve “*single-photon non-linearities*” and create nonclassical light in “*quantum nonlinear optics*” experiments.

The discussion of “*polariton condensation*” experiments that, e.g., study “*superfluidity*” and “*vortices*” in the introduction was mostly to introduce an uninitiated reader to the broad range of nonlinear optical experiments carried in the exciton-polariton systems. Indeed, as pointed out by the reviewer these are effects seen in the high-density regime where collisional interaction becomes dominant.

It is this spirit in which we have formulated the introduction to our article, and we hope that the above explanations clarify the evident perspectives of our work.

4. *“I hope that the authors do not tend to force a new interpretation on already known effects with a new „exciton blockade” which is not precise enough. The observed effects can be interpreted within the quenching of the Rabi splitting induced by the saturation of molecular optical transitions due to states filling at higher excitation densities (Pauli blocking), the saturation of exciton oscillator strength.”*

We thank the referee for bringing up this. It seems that there is some confusion here and perhaps the referee is thinking of organic semiconductors. Please, see e.g. Ref. [Yagafarov, et al., *Comm. Phys.* **3** 18 (2020)] on “*organic polariton condensates*” where “*the saturation of molecular optical transitions due to states filling at higher excitation densities (Pauli blocking)*” is discussed. Such systems of Frenkel excitons are obviously different from the present situation. In fact, the theoretical understanding (see Supplementary section of S5 [*Comm. Phys.* **3** 18 (2020)]) is based on one-body physics (e.g. a quadratic Hamiltonian for effective spins and photons) where the nonlinearity arises from the spin (or two-level) description of the excitons in full equivalence to the well known saturation (or Kerr nonlinearity) of two-level atoms. We hope that the referee can agree that the present mechanism is different, where the nonlinearity arises from interactions (be it exchange or electrostatic) and not from simple one-body saturation of two-level systems. Therefore, we believe that drawing an analogy to such settings would be not precise enough, and could potentially lead to confusion, such that we would like to refrain from attempting such comparisons.

However, we certainly agree that the “*Pauli blocking*”, in a different context, can cause an “*exciton blockade*” as we describe in our manuscript. Pauli exclusion between the Fermionic constituents of excitons leads to an exchange interaction between the excitons, which contributes to the total interaction $U(r)$ (also containing electrostatic parts). We hope that the reviewer can agree that this exchange interaction can cause collisions and collisional line shifts of emission spectra, as well as an excitation blockade that affects the reflection spectra as observed in our experiment. Therefore, the described exciton blockade is by no means a newly introduced interpretation but presents a general concept that applies to exchange interactions (Pauli blocking,) and electrostatic interactions of semiconductor excitons. The spatial blockade picture employed here has the added benefit of more clearly elucidating the spatially correlated nature of the light-matter interaction. Most importantly, the applied framework provides a conceptual simple approach to obtaining the blockade radii from our measurements and demonstrate the expected Rydberg scaling of this quantity. We, therefore, think that there are several merits to the described blockade picture, and hope that the reviewer can agree to some of them. We thank the reviewer for pointing out that the equivalence to other interpretations can be described better and have now added additional text and discussions that make the direct connection to the equivalent momentum treatment (phase-space filling) clearer [**pages 8, 9 in revised manuscript**].

5. *“Equations (1-2) refer to the same effect of the non-linear change of the refractive index due to the saturation of the excitonic transition, where at high densities the occupied states do not take part in the absorption process and reduce Rabi splitting. Authors refer to the „close analogy between the optical non-linearities in cold atomic Rydberg gases” but the reference is required as this analogy is not evident. What’s more the atomic Rydberg gases in cavities (Rabi oscillations) differ much from the polariton system (vacuum field Rabi oscillations).”*

We thank the reviewer for pointing out that a reference might be useful in this context and have now added references in the sentence in question. These include theory, experiments and overview articles that should be helpful in appreciating the close connection between atomic systems and interacting excitons. Indeed, the underlying theoretical formalism and mechanism (spatial excitation blockade) are very much analogous and both give rise to a nonlocal nonlinearity [*Phys. Rev. Lett.* **107**, 153001 (2011)]. We believe that this analogy is useful, since it shows that the effect can be quite useful, while it is often disregarded as a mere reduction of the Rabi splitting.

In fact, previous work on atomic systems that showed the very same blockade effect that leads to a seeming reduction of the polariton splitting [*Phys. Rev. Lett.* **105**, 193603 (2010)] can be exploited to generate strong single-photon interactions [*Nature* **488**, 57 (2012)]. It now appears that this important point about the nature and application of the reported nonlinearities is beginning to be recognized in semiconductor context [see e.g. *Phys. Rev. Lett.* **125**, 197402 (2020)] for similar arguments related to trions). We therefore believe that the employed blockade picture and its link to atomic systems provides a fruitful analogy with regards to future applications and recognizing the potential of this type of nonlinearity (in the regime of low densities where the shift scale linearly with intensity).

We would, finally, like to point out that this analogy of the underlying nonlinear mechanism holds irrespective of the light-matter coupling, e.g. whether one uses cavities, free-space coupling or multi-level driving. Moreover, we would like to clarify that all recent Rydberg-atom optics experiments have been performed in the strong-coupling regime, in fact, much stronger than what is typically possible in semiconductor settings. For example, free-space experiments with slow-light Rydberg polaritons achieve an energy splitting between the polariton branches (dark and bright state polaritons) that are 3 orders of magnitudes larger than the natural linewidth of the low-lying atomic state, and more importantly 6-7 orders of magnitude larger than the excited Rydberg state. Therefore, these experiments achieve remarkably large energy splitting between polaritons (vacuum) and not just classical-field Rabi oscillations.

6. *“Despite the above criticism, I have to admit that the results are very valuable from the point of view of recent publications. In my opinion the authors should put more emphasis on the observed saturation densities which are relatively low. With comparison to V. Kravtsov et al. Light Sci. Appl. 9, 56 (2020) where the Mott transition is claimed to be at $10^{14}/\text{cm}^2$ and the polariton density at $10^{12}/\text{cm}^2$, in the manuscript authors have the densities up to at $10^{11}/\text{cm}^2$ and already observe the Rabi quenching. This demonstrate huge discrepancy in the literature.”*

There is indeed a discrepancy between the numbers in the paper by *Kravtsov et al.* where they obtain $g_X \sim 1\mu\text{eV}\mu\text{m}^2$. A similar report (Stepanov et al. ArXiv2007.00431) shows the role of the saturation nonlinearity and the exchange interaction where the combined nonlinear interaction strength is $\sim 7\mu\text{eV}\mu\text{m}^2$. This is close to the values we have ($g_{\text{pol-pol}}^{1s} \sim 10.0 \pm 4.2\mu\text{eV}\mu\text{m}^2$).

The reduction of the Rabi splitting on the scale we observe is therefore consistent with the theoretical expectation (using Eq. (2) as an estimate or the k-space PSF approach of Schmitt-Rink et al., *Phys. Rev. B* **32**, 6601–6609 (1985)). Please, note that the experiment of [*Light Sci. Appl.* **9**, 56 (2020)], does not present measurements of the nonlinear shift of the upper polariton branch that would substantiate the claimed density of 10^{14}cm^{-2} . In fact, the only other measurement that shows the nonlinear behavior of both polariton branches is a very recent work (Stepanov et al.

ArXiv2007.00431) which agrees with our numbers for the 1s state. Our work showing the PSF effect for 1s and 2s excitonic states in TMDs is to the best of our knowledge, the first to demonstrate this clearly. Besides showing that exciton blockade or PSF dominates in TMD microcavities, the main result is, however, the enhancement of the nonlinearity for excited state excitons.

7. “*Summarizing my remarks, the novelty of the research is not very high:*
- *the observation of exciton polaritons in 2s state in the absorption process in 2D materials is not new [M. Krol et al 2020 2D Mater. 7 015006]*
- *the model is also widely used to describe the saturation of molecular optical transitions due to states filling at higher excitation densities*
- *the idea of admixture of higher excited states to polariton modes to increase the non-linear effects was also demonstrated [V. Walther Nature Communications 9, 1309 (2018); V. Kravtsov et al. Light Sci. Appl. 9, 56 (2020)], also with 2p that have a dipole moment [Nano Lett. 2020, 20, 1676–1685]”*

We respectfully disagree with this opinion of the reviewer. The important novel results of our work are the realization of strong cavity-coupling to excited-state excitons in TMD materials and the demonstration of enhanced optical nonlinearities of the thereby formed polaritons. The reviewer bases his/her opinion about the novelty of these results on some recent references, which we would like to discuss one by one in order to prove and substantiate our point:

- *“the observation of exciton polaritons in 2s state in the absorption process in 2D materials is not new [M. Krol et al 2020 2D Mater. 7 015006]”*

As we describe in our manuscript, Rydberg states have indeed been observed in semiconductors, including TMDs, and we have already given corresponding references. Importantly, however, the realization of strong cavity coupling and the formation of polaritons with excited-state excitons in a two-dimensional material could not be achieved so far, and certainly not in the above reference. On the contrary, the authors of that article explicitly write (on page 5)

“It is worth to mention that due to low oscillator strength of the excited 2s exciton state in our samples, we do not observe a strong coupling regime at this transition, as shown in figure S9.”

This has now been achieved for the first time in our work. In the same paragraph, the authors motivate this goal by stating that

“Such a coupling with excited exciton states has recently become of interest due to large optical nonlinearities one can probe in this type of systems”,

and, in fact, refer to our recent theory work [*Nature Comm.* 9, 1309 (2018)] (see below), where we outline the potential of excited-state interactions to enhance optical nonlinearities.

- *“the model is also widely used to describe the saturation of molecular optical transitions due to states filling at higher excitation densities”*

As described above, this is not the case. *“The saturation of molecular optical transitions”*, referred to by the reviewer, is modeled (see Supplementary section S5 of [Comm. Phys. 3 18 (2020)]) as saturation of two-level systems that are used to describe Frenkel excitons in organic semiconductors. This kind of modeling is different from the present framework which is based on interactions.

However, we see it as a strength of the present treatment that it is indeed of broader applicability and has been applied to other situations, which we cite in the manuscript. Importantly, this includes blockade due to electrostatic interactions between highly-excited Rydberg excitons [*Phys. Rev. Lett.* **125**, 097401 (2020)] and atoms [*Phys. Rev. Lett.* **107**, 153001 (2011), *Advances in Atomic, Molecular, and Optical Physics* **65**, 321 (2016), *Nature Photonics* **8**, 685 (2014)], as well as exchange interactions between excitons [*Phys. Rev. B* **61**, 13856 (2000)].

Yet, these conceptual links to other systems should not take away from the novelty of our core results, namely the realization of strong cavity-coupling to excited-state excitons in TMD materials and the demonstration of enhanced optical nonlinearities of the thereby formed polaritons.

- “*the idea of admixture of higher excited states to polariton modes to increase the non-linear effects was also demonstrated [V. Walther Nature Communications 9, 1309 (2018); V. Kravtsov et al. Light Sci. Appl. 9, 56 (2020)]*”

First note that [V. Walther *Nature Communications* **9**, 1309 (2018)] is a theory work by two of the authors of the present manuscript, where the potential for enhancing excitonic nonlinearities via excited states has been pointed out. In fact, these predictions motivated us to team up and demonstrate these ideas for the first time in experiments with TMD cavities.

Ref. [V. Kravtsov et al. *Light Sci. Appl.* **9**, 56 (2020)], on the other hand, does not report nonlinearities for excited exciton states (only for 1s), in contrast to what the reviewer seems to suggest.

Therefore, none of these references affect the novelty of our main results, namely the realization of strong cavity-coupling to excited-state excitons in TMD materials and the demonstration of enhanced optical nonlinearities of the thereby formed polaritons.

- “*the idea of admixture of higher excited states to polariton modes to increase the non-linear effects was also demonstrated [...] with 2p that have a dipole moment [Nano Lett. 2020, 20, 1676–1685]*”

Ref. [*Nano Lett.* **20**, 1676 (2020)] used 2p excitons to study two-photon processes, such as two-photon photoluminescence excitation and second-harmonic generation spectroscopy, as clearly stated in that article. These are higher order optical processes that are based on single-exciton physics, and do not involve polariton nonlinearities based on exciton interactions. In particular, they do not rely on the dipole moment or dipole-dipole interaction of the 2p state, in contrast to what the reviewer seems to imply.

Therefore, this reference does not relate to our main results, namely the realization of strong cavity-coupling to excited-state excitons in TMD materials and the demonstration of enhanced optical nonlinearities of the thereby formed polaritons.

Altogether, this brief discussion clarifies that none of the provided references affects the novelty of our results, namely the realization of strong cavity-coupling to excited-state excitons in TMD materials and the demonstration of enhanced optical nonlinearities of the thereby formed

polaritons. On the contrary, some of the references even mention these points as important milestones, which have now been reached in the present work.

8. *“the interpretation of the observed effects of the quenching of Rabi splitting should be corrected and the different types of interactions in polariton system should be clarified and properly addressed.”*

As clarified above, the suggested analogy to molecular saturation of Frenkel excitons (see, e.g., [Comm. Phys. **3** 18 (2020)]) does not apply. In fact, the underlying theoretical picture/model (see, e.g., Supplementary section of S5 [Comm. Phys. **3** 18 (2020)]) would not yield an increased nonlinearity for excited states, since there the nonlinearity arises from one-body saturation due to the two-level description of the exciton in terms of spin operators, which saturate irrespective of the actual exciton state. This clearly shows that drawing such an analogy or using such an interpretation would be incorrect for our experiment.

To address the reviewer’s comments, we have, however, added a dedicated section that now better highlights the close relation/equivalence of the spatial exciton blockade picture and the k-space treatment of PSF [**pages 8, 9 in revised manuscript**]. Both yield a consistent interpretation of our results, and we believe that pointing out this connection and ensuing analogies between atomic and semiconductor settings better reveals the potential of the observed nonlinear behavior, which is known to go beyond a mere reduction of the Rabi splitting for classical light.

Once again, we are happy that the reviewer finds our “*experiment of importance*” and would like to thank the reviewer for concluding that its “*result is very valuable*”. At the same time, we hope that the above clarifications and revisions to the manuscript now make a better point for its novelty and implications, to allow the reviewer to reach a positive conclusion about publication of our manuscript in Nature Communications.

REVIEWERS' COMMENTS

Reviewer #1 (Remarks to the Author):

I thank the authors for their effort to thoroughly address all of my comments. I have also read the author's responses to other Reviewers' comments and in my opinion, the authors have managed to address all the raised questions and comments. I thus believe that the work can be accepted for the publication in Nature Communications.

Reviewer #2 (Remarks to the Author):

I reviewed the revised manuscript. I think both of my the other reviewers' questions were properly addressed. Therefore I support its publication.

Reviewer #3 (Remarks to the Author):

I would like to thank the Authors for detailed answers to my comments. I read them in detail, together with all changes in the manuscript. In my opinion, the manuscript has changed a lot and now gives a clear explanation of the observed effects. I'm pleased to admit that the manuscript is much improved and the description is much better. The authors provide very convincing arguments in response to all questions raised by the Referees.

I'm still not convinced that the references to condensation, superfluidity and vortices are appropriate, rather confusing, but the rest of the manuscript now merits a publication in Nature Communication. The current description of the nonlinearity and the model are of the highest quality in the manuscript.